# Heart girth best predicts live weights of market-age pigs in Tanzania

**Mwemezi L. Kabululu**(ID)*

Tanzania Livestock Research Institute (TALIRI), Central Zone Office, Mpwapwa, Dodoma, Tanzania

* mwemezie@gmail.com

## Abstract

The aim of this study was to use linear body measurements to develop and validate a regression-based model for prediction of live weights (LW) of pigs reared under smallholder settings in rural areas in the southern highlands of Tanzania. LW of 400 pigs (range 7 to 91 kg) was measured, along with their heart girths (HG) and body lengths (BL). BL was measured from the midpoint between the ears to the tail base. HG was measured as chest circumference just behind the front legs. LW was determined using a portable hanging scale. An analysis of covariance was performed to test for differences in LW between male and female pigs, including age, HG and BL as covariates. LW was regressed on HG and BL using simple and multiple linear regressions. Models were developed for all pig ages, and separately for market/breeding-age pigs and those below market/breeding age. Model validation was done using a split-samples approach, followed by *PRESS*-related statistics. Model efficiency and accuracy were assessed using the coefficient of determination, $R^2$, and standard deviation of the random error, respectively. Model stability was determined by assessing 'shrinkage' of $R^2$ value. Results showed that HG was the best predictor of LW in market/breeding-age pigs (model equation: LW = 1.22HG—52.384; $R^2$ = 0.94, error = 3.7). BL, age and sex of pigs did not influence LW estimates. It is expected that LW estimation tools will be developed to enable more accurate estimation of LW in the pig value chain in the area.

## 1. Introduction

In Tanzania, pig/pork production and marketing have continued to record a remarkable growth over recent decades, as compared to other livestock sectors, contributing substantially to improving livelihoods of rural and urban communities [1, 2]. Most (>90%) of the pigs in Tanzania are produced in low-input output systems, largely in rural areas, involving smallholder backyard units with herd sizes of usually less than 10 pigs [1].

Pig marketing in many areas of Tanzania largely involves farm-gate transactions between a pig trader and a farmer [3]. Traders usually buy pigs based on their subjective visual estimates of weight, in most cases to their advantage by underestimation (Walugembe et al., 2014) [4]. This results in farmers receiving less money than they would have rightfully deserved. This could be addressed by valuing and selling/buying pigs on the basis of price/kg of live weight

**Data Availability Statement:** The dataset generated and analyzed during the current study has been provided as a supplementary file.

**Funding:** The author received no specific funding for this work.

**Competing interests:** The author has declared that no competing interests exist.

(LW). Apart from its usefulness in pig marketing, estimation of pig weight is important for optimization of management, performance and hence profitability [5, 6]. It enables accurate calculation of feed requirements [7, 8]; monitoring of pig growth and health status; and determination of correct drug dosages [9, 10]. Therefore, the importance of having optimally accurate and affordable methods for estimating pig weights cannot be overemphasized.

There are two methods which can be used to estimate pig weights, direct and indirect methods [11]. A direct method involves the use of calibrated electronic or mechanical weigh scales, while indirect methods include visual estimation of weights; imaging and use of linear body measurements. Despite being an accurate and widely accepted method globally, the use of weigh scales is limited in most smallholder pig farming settings, Tanzania inclusive. Apart from inaccessibility, their use requires high input of time and labour, and can cause stress and injury to pigs and handlers [12, 13]. Another drawback is the lack of technical skills in operating and maintaining the scales [14, 15]. Due to the limitations, indirect methods of weight estimation present a better option, with the use of body measurements being the most common [11]. Body measurements provide a relatively accurate, consistent and cheaper method of estimating LW.

Several studies in different countries have described the relationship between the linear body measurements and pig weight, and developed weight prediction models [4, 12, 16–18]. However, due to differences in management practices, and in pig body sizes and conformations, weight estimation methods developed in one locality may not be suitable for use in another locality [6, 19]. Hence, for optimal accuracy, weight estimation methods should only be applied to pigs reared under similar conditions of management, and more or less of the same genotype.

Therefore, this study had two objectives. Firstly, the study aimed to develop and validate a regression-based equation which can be used to accurately estimate LW of pigs reared by smallholder farmers in rural areas of the southern highlands of Tanzania, by using simple linear body measurements, in this case, heart girth (HG) and body length (BL). In addition, this study also aimed to validate use of a commercially available calibrated heart girth tape for use on market-age pigs in the area, by assessing difference in mean between actual weight measured using a hanging scale and weight estimated by the heart girth tape.

## 2. Materials and methods

### 2.1 Statement of ethical consideration

The conduct for this study fitted within the conduct of the field intervention trial which was approved by the Research, Publications and Ethics committee of the College of Veterinary Medicine and Biomedical Sciences (CVMBS) of the Sokoine University of Agriculture, SUA, in Tanzania (Reference number: SUA/CVMS/016/32). All participating farmers provided written informed consent to participate. The trial was conducted as guided by the principles of good clinical practice. Animal handling adhered to Animal welfare regulation stipulated in section 40 of the Tanzania's Animal Welfare Act number 19 of 2008. Animals were handled by livestock technicians who also received pre-trial training on best practices in handling animals.

### 2.2 Study sites and study animals

The study was conducted in 10 villages from Mbeya rural and Mbozi districts located in the southern highlands zone, south western Tanzania. The two districts are, to a great extent, rural and pig rearing was largely on a small scale, with majority of farmers keeping one to five pigs [2]. Most of the pigs were nondescript, being crossbreeds of indigenous breeds and exotic breeds mainly Landrace and Large white [20, 21]. Pig rearing practices were rudimentary. A

study by Kabululu et al. [21] showed that about 60% of pigs are kept in pens while 20% are left to scavenge for food. Constructed pens had either raised floors (slatted with timber off cuts or with tree logs/sticks) or ground floor (earthed or with concrete). The pen walls had either timber off cuts, tree poles/sticks, bamboo poles or bricks. Roofing materials used were grass/thatches, bamboo poles or iron sheets.

Maize bran and kitchen wastes were the main feeds provided to pigs with little or no protein or mineral supplementation. Selling/buying of pigs relied solely on visual estimates of body size/weight of pigs with virtually no actual weighing of pigs.

## 2.3 Data collection

Data were collected during household visits in June to July 2017. Weights and body measurements were recorded from 400 pigs, excluding sick, pregnant and nursing sows. From the list of all households whose heads had consented to participate in the main (intervention) study [22], pig measurements (LW, HG and BL) were taken from a maximum of two pigs, in every other household that was visited. To ensure accurate measurements, pigs were restrained and their heads stabilized by using a pig snare. HG and BL were measured by using ANIMETER®, a calibrated commercial weigh tape (BA 219 Animeter 050, Albert Kerbl GmbH, Buchbach, Germany). BL, measured on the back of the animal, was defined as the length (in centimeters) from the midpoint between the ears to the base of the tail [4]. HG, also in centimetres, was defined as the chest circumference, measured by wrapping the measuring tape snuggly around the chest area, just behind the front legs. Pig weighing was done using a portable hanging scale (with a precision of 500 g). Pig sex and age (in months) were also recorded. Estimation pig age was done by pig owners, and by confirmation by using available records.

For validation of a HG tape (ANIMETER®), LWs of a random sample of 40 pigs (from a sampling frame of the 400 pigs) were estimated using the tape, in addition to weights measured by the hanging scale. This was achieved by including every 10th pig, from all the pigs that were measured. The tape had values of estimates of LW printed on it, corresponding to measured HG values.

## 2.4 Data analysis

Analysis of data was done using STATA© (Stata Corporation, College Station, Texas) version 14. Descriptive statistics were used to determine means (± SD), medians, minimum and maximum values, and 25% and 75% percentiles of different parameters. Pearson pairwise correlation analysis was performed to determine the degree of a linear relationship between body measurements and LW.

A preliminary analysis of covariance (ANCOVA) was performed to test for differences in LW between males and females; including age, HG and BL as covariates included as main effects and as interaction terms of Age*HG and Age*BL.

LW was regressed on HG, BL and age using simple and multiple linear regressions. The general linear equation used was;

$$Y = \beta_0 + \beta_i X_i + \varepsilon,$$

where;

Y = Live (body) weight
$\beta_0$ = constant (intercept)
βi = regression coefficient of the ith independent variable (slope of the regression line)
Xi = The value of the ith independent variable
ε = residual (error term)

Regression analysis was performed in two stages, first by including all pigs and second by categorizing pigs into those below 12 months (below market/breeding age), and those 12 months and above (market/breeding-age pigs). In the first stage, a multiple regression analysis was performed regressing LW on HG and age as main and interaction effects; and simple regression analyses regressing LW on each of the two variables (HG and age) separately. In the second stage, a simple regression of LW on HG was performed for each of the two age categories.

The resulting regression models were tested for normality and homoscedasticity of residuals, both visually and computationally. Normality was tested using a Shapiro-Wilk test. Homoscedasticity, an assumption of constant variances of the residuals was determined by assessing scatter plots of residuals against fitted values and using Breusch-Pagan/Cook-Weisberg test.

Model efficiency was tested using the coefficient of determination (goodness of fit, $R^2$), the percentage of the variance in the LW explained by predictors. Model accuracy was estimated by assessing root mean square error (RMSE), which is the standard deviation of the random error.

Further, the accuracy of the more predictive model was determined from residual statistics. To explain distribution of the residuals, mean, median, 25% and 75% percentiles were determined. Mean absolute value of residuals was used to determine average prediction error. Prediction intervals and confidence intervals around mean predicted weight were determined using lower and upper bounds of predicted values and standard errors of forecast and prediction, respectively.

For model cross-validation, a split-samples approach was used, followed by PRESS-related statistics. A random selection of 75% of all pigs (model derivation group) was used to develop the model and the remaining 25% (model validation group) was used for model validation [18, 23, 24]. PRESS-related statistics involved the use of PRESS, which is the predicted residual sum of squares. The more predictive model was used to estimate weights of the pigs in the validation group, and PRESS was determined from the calculated residuals. Then the PRESS was used to calculate a $R^2$ PRESS, a modified form of $R^2$, using the formula: $R^2$ PRESS = 1 –[PRESS /SS total], where SS total equals the sum of squares for the original regression equation [25, 26].

Model stability (hence reliability) was then assessed by estimating 'shrinkage', which was the difference between $R^2$ and $R^2$ PRESS. The smaller the 'shrinkage', the more stable the model was considered to be.

The difference in mean between actual weight measured by the scale and weight estimated by the heart girth tape was compared using a paired t test. Differences between measured weights and weights estimated using the tape were considered as residuals, and these were compared to the differences between measured weights and weights predicted by the model. Difference in mean absolute residuals, together with 25% and 75% percentiles were determined.

## 3. Results

Among the 400 pigs included in the study, 242 (60.8%) were females and 156 (39.2%) were males. Age, LW, HG and BL all measured pigs are reported in Table 1. Two pigs had missing age information. Age ranged from four to 44 months, with a mean of 10 months and a median of 8 months. When categorized into age groups, 299 (74.7%) were those below market/breeding age (below 12 months) and 101 (25.3%) were market/breeding-age pigs. Mean live weight was 28.4 kg, with a mean of 24.3 kg for pigs below market age and 40.3 kg for market/breeding-age pigs.

**Table 1. Means, standard deviations (SD), minimum, maximum, range of live weights, age, and body measurements (heart girth and body length) of 398 pigs (242 females, 156 males) pigs reared by smallholder farmers in Mbeya rural and Mbozi districts, Tanzania.**

| Parameter | Sex* | Mean | SD | Minimum | Maximum |
|---|---|---|---|---|---|
| Age (months) | Males | 8.6 | 3.7 | 4 | 32 |
| | Females | 10.9 | 5.6 | 4 | 44 |
| | All pigs | 10.1 | 5.1 | 4 | 44 |
| Live weight (kg) | Males | 25.2 | 9.6 | 8 | 60 |
| | Females | 30.5 | 13.7 | 7 | 91 |
| | All pigs | 28.4 | 12.5 | 7 | 91 |
| Heart girth (cm) | Males | 62.7 | 9.3 | 38 | 91 |
| | Females | 67.8 | 11.9 | 41 | 118 |
| | All pigs | 65.7 | 11.3 | 38 | 118 |
| Body length (cm) | Males | 73.8 | 12.1 | 45 | 103 |
| | Females | 80.1 | 14.5 | 43 | 137 |
| | All pigs | 77.6 | 14 | 43 | 137 |

* sex of two pigs was not recorded

Results of a pairwise correlation analysis (Table 2) showed that, overall, HG had the highest correlation with LW (r = 93%) while age and LW were moderately correlated (r = 59%). Age was also moderately correlated with HG and BL. Stratified by age groups, higher correlations were found in pigs aged 12 months and above.

The analysis of variance (Table 3) showed that pig sex had no effect on LW (p = 0.571). HG showed a significant effect on LW as both main and interaction effects were significant. Age also had a significant effect on LW, both as a main effect and as an interaction with HG. However, BL had no significant relationship with LW, hence it was not included in the subsequent regression modelling.

Results of the regression analysis (Table 4) showed that the model regressing LW on HG in market/breeding-age pigs (equation 5: LW = 1.222HG—53.384 + $\varepsilon$) was the best model. This means HG was best correlated with LW in market/breeding-age pigs ($R^2$ = 93.8%) and prediction of LW from HG had the lowest prediction error (RMSE = 3.7). Age alone was not reliable in the prediction of LW ($R^2$ = 32.9%).

**Table 2. Pearson correlation coefficients (*r*) for correlation between live weight and body measurements (heart girth and body length) of 400 pigs reared by smallholder farmers in Mbeya rural and Mbozi districts, Tanzania.**

| | Parameter | Live weight | Heart girth | Body length | Age |
|---|---|---|---|---|---|
| **All pigs (n = 400)** | Live weight | - | - | - | - |
| | Heart girth | 0.93* | - | - | - |
| | Body length | 0.86 | 0.89 | - | - |
| | Age | 0.59 | 0.56 | 0.54 | - |
| **Pigs below market/breeding age (n = 299)** | Live weight | - | - | - | - |
| | Heart girth | 0.89 | - | - | - |
| | Body length | 0.79 | 0.84 | - | - |
| **Market/breeding-age pigs (n = 101)** | Live weight | - | - | - | - |
| | Heart girth | 0.96 | - | - | - |
| | Body length | 0.86 | 0.86 | - | - |

*All correlations were significant (p = 0.000)

**Table 3. Analysis of variance of the difference in live weights between male and female pigs including age, heart girth and body length as covariates.** Data were collected from 400 pigs reared by smallholder farmers in Mbeya rural and Mbozi districts, Tanzania.

| Source | Partial SS | df | MS | F-value | P-value |
|---|---|---|---|---|---|
| **Model** | 55,398.9 | 6 | 9,233.2 | 565.5 | 0.000 |
| **Sex** | 5.2 | 1 | 5.2 | 0.32 | 0.571 |
| **HG** | 861.3 | 1 | 861.3 | 52.8 | 0.000 |
| **BL** | 17.6 | 1 | 17.6 | 1.1 | 0.300 |
| **Age** | 676.1 | 1 | 676.1 | 41.3 | 0.000 |
| **HG*Age** | 91.6 | 1 | 91.6 | 5.6 | 0.018 |
| **BL*Age** | 11.7 | 1 | 11.7 | 0.7 | 0.397 |

*SS = sum of squares; df = degrees of freedom; MS = mean squares

Homoscedasticity was verified for the model no. 5 (p = 0.39, $X^2$ = 0.71), while it was not for the remaining models (equations 1–4). The normality test showed that residuals for the model equation no. 5 were normally distributed (p = 0.197, Z value = 0.853), while residuals for the remaining models (1 to 4) were not normally distributed.

Fig 1 shows residual-versus-fitted plots following the regression analyses. In each of the plots A–D, corresponding to model equations 1–4, a pattern is observed, indicating violation of the least squares assumption. In the plots, an increase is observed in the variation of the residuals as the fitted values increase (heteroscedasticity). In addition, in plots A, B and D, the residuals show a curvature, indicating a violation of the assumption of linearity; while for plot C, there is skewing of the values of data points. On the other hand, plot E which corresponds to model equation number 5, shows a random and uniform scatter of points (residuals), indicating a good fit.

Fig 2 shows line of best fit for the more predictive model, together with 95% confidence and prediction intervals. The confidence interval (shaded area around the best-fit line) shows the area where there is a 95% probability that true best-fit line lies. The prediction interval (whole area between the two outer lines) shows the 95% probability that weights predicted by the model will lie within this interval range.

"Shrinkage" was determined to be relatively low (0.031), and according to suggestion by Kleibaum et al. [27] (shrinkage <0.1 = stable model), the more predictive model (model 5) was considered to be stable and hence was validated. The regression model results indicated that weight increased by 1.22 kg as heart girth increased by 1 cm. The model yielded a prediction error of 3.7 kg while the mean predicted weight was 39.7 kg. This translates to 9.3% of the

**Table 4. Results of the regression analysis of live body weight on heart girth and age of different categories of 400 pigs reared by smallholder farmers in Mbeya rural and Mbozi districts, Tanzania.**

| Pig group | Parameter | Model No. | Regression equation | $R^2$ | adj. $R^2$ | RMSE (ε) | F value* |
|---|---|---|---|---|---|---|---|
| All pigs | HG, Age | 1 | LW = 0.989HG + 0.252BL − 39.237 ± ε | 86.4 | 86.3 | 4.7 | 925.95 |
| | HG | 2 | LW = 1.053HG − 40.931 ± ε | 85.6 | 85.6 | 4.8 | 1748.79 |
| | Age | 3 | LW = 1.400Age + 15.062 ± ε | 32.9 | 32.6 | 10.4 | 145.82 |
| Pigs below market/breeding age | HG | 4 | LW = 0.807HG − 26.931 ± ε | 74.9 | 74.8 | 4.2 | 656.37 |
| Market/breeding-age pigs | HG | 5 | LW = 1.222HG − 52.834 ± ε | 93.8 | 93.7 | 3.7 | 1108.79 |

HG = heart girth; BL = body length; LW = live weight; $R^2$ = coefficient of determination; adj. $R^2$ = adjusted $R^2$; RMSE = root mean square of error

*all prediction equations were significant (p<0.05)

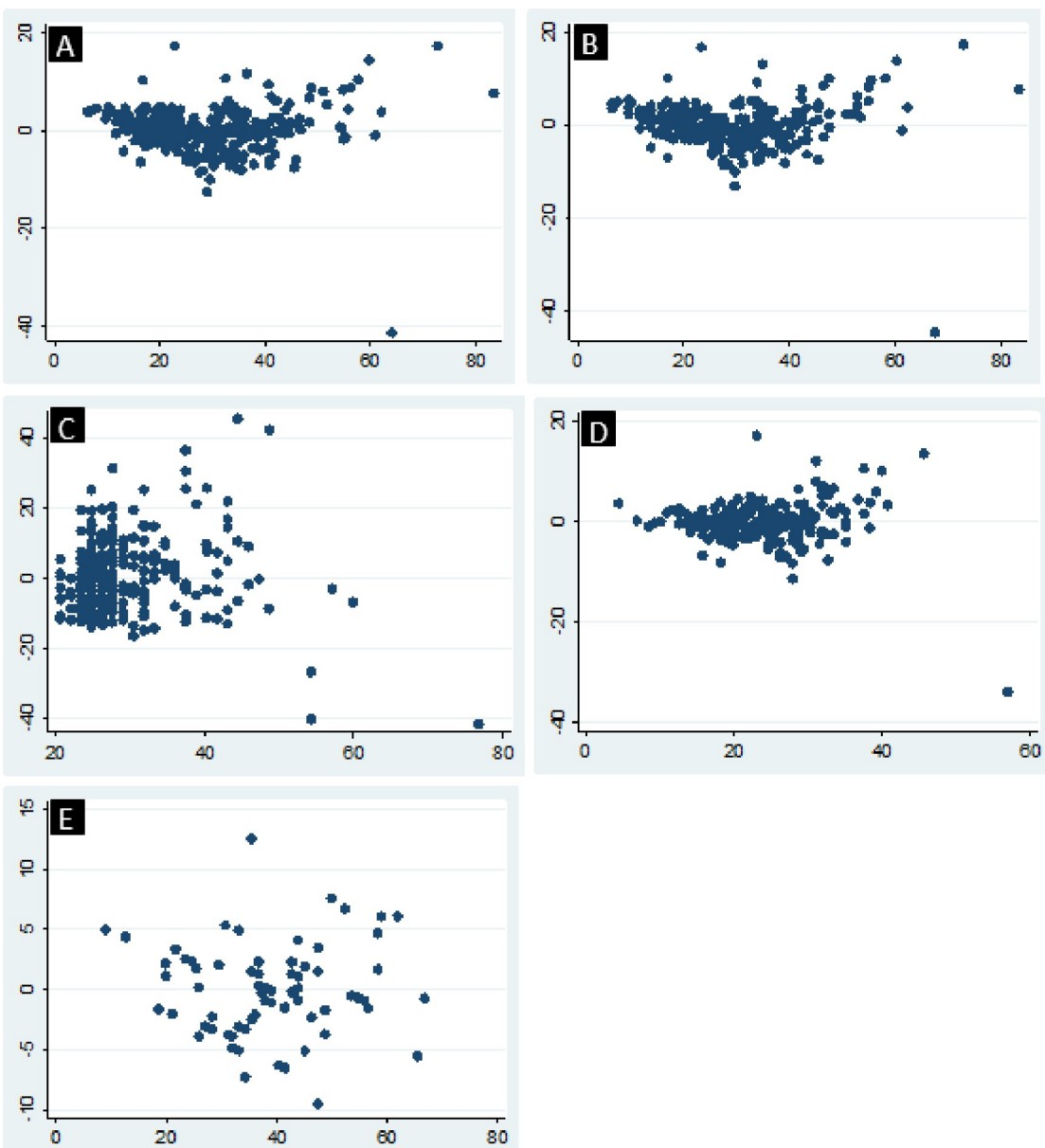

**Fig 1. Plots of residuals (Y axis) versus fitted values (X axis) following regression of live weight (LW) on heart girth (HG) and age of 400 pigs reared by smallholder farmers in Mbeya rural and Mbozi districts, Tanzania.** A = regression of LW on HG and age (pigs of all ages); B = regression of LW on HG (pigs of all ages); C = regression of LW on age (pigs of all ages); D = regression of LW on HG (pigs below 12 months); E = regression of LW on HG (pigs 12 months and above).

predicted LW. Mean residual was close to zero (-5.17e-09) indicating a good fit of the regression line. The model will predict weight with a mean error of $\leq 0.19$ kg for half of the pigs and a mean error of $\leq 2.03$ kg for 75% of the pigs.

Paired t test showed that there was no significant difference in mean between weights measured by the hanging scale and weights estimated using the heart girth tape. Nevertheless, mean absolute residual was higher for weight estimation using the heart girth tape as compared to estimation using the model (Table 5). In addition, although the median values and

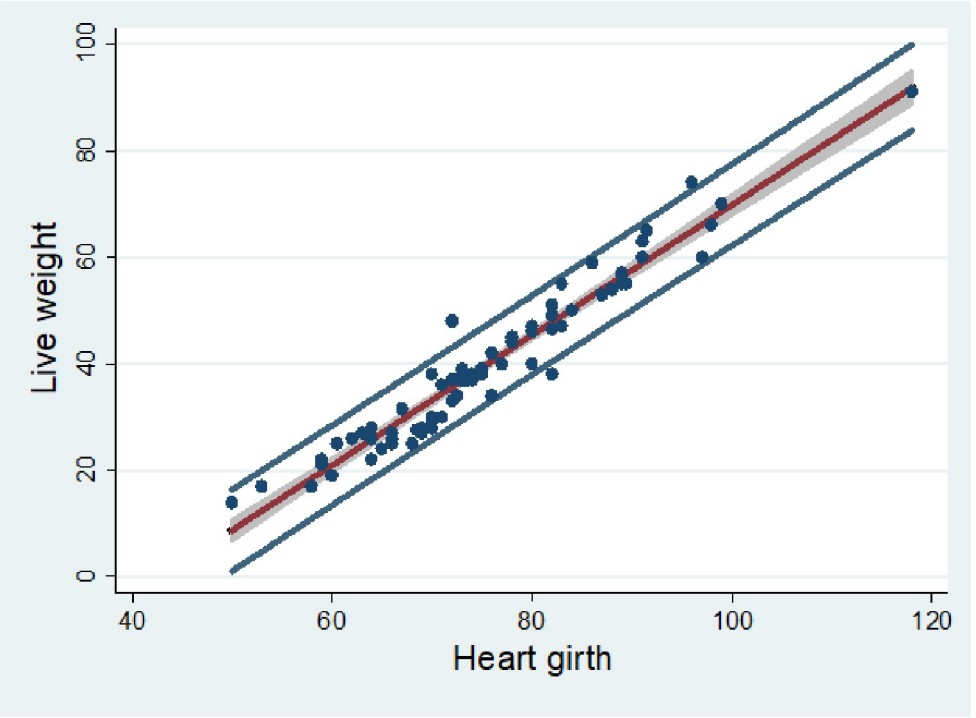

**Fig 2. Scatter plot of live weight vs heart girth showing line of best fit, overlaid with 95% confidence interval (shaded area around the line of best fit) and prediction intervals (area between outer lines).**

25% percentiles were comparable, 75% percentile was higher for residuals from weight estimation using the heart girth tape than for weight estimation using the model.

## 4. Discussion

In Tanzania, there is a paucity of studies which assessed correlation between body weight and linear body measurements of livestock. This study could be the first to determine the relationship between linear body measurements and body weights in Tanzanian pigs. The study has shown a strong correlation ($R^2 = 0.94$) of HG and LW of market/breeding-age pigs in a major pig production area in the country. Hence, HG could be reliably used for estimation of LW of market/breeding-age pigs in the area, with relatively high accuracy.

HG is probably the most commonly used body measurement parameter to predict LW in livestock. A number of studies have reported that HG is best correlated with LW in pigs, when

**Table 5. Mean absolute differences between measured (live) pig weight and weight estimated by heart girth tape for a random sample of 40 pigs reared by smallholder farmers in Mbeya rural and Mbozi districts.**

| Parameter | Residuals (weight predicted by the model) | Residuals (weight estimated by the heart girth tape) |
|---|---|---|
| Mean | 3.03 | 3.49 |
| Standard deviation | 2.49 | 2.80 |
| Median | 1.99 | 2.00 |
| 25% percentile | 1.17 | 1.00 |
| 75% percentile | 4.78 | 6.00 |

used as a single predictor. In Ugandan pigs, HG has been reported to be the most predictive single body measurement, with an $R^2$ value of up to 0.94 [4, 28]. In Zimbabwe, Sungirai et al. [19] reported a coefficient of determination of 0.89 for relationship between HG and LW in Large White and Landrace breeds. Machebe and Ezekwe [17] studied growing to finishing pigs in Nigeria and found that HG was highly correlated ($R^2$ = 0.98) with LW. In Ghana, HG yielded the highest prediction ability (Adjusted $R^2$ = 0.77 to 0.92) in large white pigs of five to 20 weeks [29]. In India, HG had the best correlation with LW of Landly pigs 9–15 months of age ($R^2$ = 0.80) [13]. Studying Philippine native pigs weighing 20–50 kg, Paras et al. [30] found that HG explained 88% (i.e. $R^2$ = 0.88) of the total variability in LW. In Ireland, O'Connell et al. [31] found that using HG alone, sow weight could be estimated with an (adjusted) $R^2$ of 0.81. Groesbeck et al. [12] reported a strong correlation ($R^2$ = 0.98) in grower to finisher pigs in the United States.

Other studies had also reported a strong LW prediction ability when HG was combined with other body measurements. Mutua et al. [18] developed a single model combining HG and BL for predicting LW in pigs in rural western Kenya, and found that the model explained 88% to 91% of the total variation in LW for pigs from $\leq$ 5 months to breeding age ($\geq$ 10 months). In Nigeria, Alenyoregue et al. [32] came up with a prediction equation with age of the pig, HG, and BL ($R^2$ = 0.93). In the study by Kumari et al. [13], including BL and a quadratic term of HG led to $R^2$ value of 82.9% compared to values of 58% and 80% when LW was separately regressed on BL and HG, respectively. Paras et al. [30] found that including BL as a predictor increased $R^2$ from 88% to 92%.

In this study, BL did not seem to significantly influence LW estimations, contrary to some previous studies. The reason for this could be the fact that HG largely measures muscle and fat tissues, which more closely reflect body condition (and hence LW) as opposed to BL which measures skeletal tissues [32]. The fact that HG could be used to predict LW with relatively high efficiency and accuracy makes it simple to develop LW estimation tools. In addition, HG has been considered to be relatively easier to measure under field conditions, as compared to other measurements [6].

HG has also been reported to be a significant predictor of LW in other types of livestock, such as cattle [15, 33, 34] and goats [6, 35, 36].

Previous studies have shown that age can be used to estimate LW of pigs [7, 19]. In this study, however, age was not found to influence LW. This can be explained by two reasons. Firstly, as the pigs in the area are mostly non-descript/ crossbred pigs, growth pattern can be expected to be different. Secondly, pig management practices in the area are generally rudimentary [21]. As a result, pigs cannot exhibit their full growth potential related to their age, and a sizeable number of pigs appeared smaller in size than their age. In addition, difference in pig management practices across households could be among reasons, as it was pointed out in Uganda [37, 38] A previous study in the area of the current study, by Kimbi et al. [3], reported that pig age was a poor determinant of price ($R^2$ = 0.22) especially for older pigs.

Although age itself was not a good predictor of LW, the observed significant interaction of HG and age indicates that age specific models could be more appropriate, as it was suggested by Mutua et al. [18]. This is more so considering the fact that HG was a significant predictor only in pigs one year and above. Further studies are suggested, including more body measurements/parameters in order to develop models for prediction of LW in younger pigs in the area.

The results of this study have shown that sex did not influence LW estimation. This is in line with previous studies in Kenya [18], Uganda [4, 28] and Zimbabwe [19]. As pointed out

before, the reason for this could be related to the pig management constraints which make the pigs unable to express their genetic potential.

Under an ideal situation, accuracy of LW estimation depends on the purpose of weight estimation. The accuracy range found in this study (9.3%) can be considered to be acceptable for the purpose of determining treatment doses [39]. In addition, it is likely to lead to more reliable weight estimates during pig selling/buying compared to visual estimates.

The heart girth tape assessed in this study was found to have a reasonable ability to estimate LW in market/breeding-age pigs in the area. It is assumed that the tape was developed for use with more exotic, well-managed pigs of Europe and North America [18]. Therefore, it was not expected to accurately estimate LW of pigs in the study area. Thus, the tape can reliably be used to estimate LW of pigs in the area until more efficient and accurate tools for weight estimation are developed.

There are a number of ways the results of this study could be put to use. One way is to develop a measuring tape which has LW corresponding to HG values printed on it. Another approach is to develop a smart phone application where farmers can input HG values and readily get LW estimates. Lastly, HG and their corresponding LW can be produced in a paper-based chart which can be made available to farmers.

## 5. Conclusion

In conclusion, this study has developed and validated a LW prediction model (LW = 1.22*2HG—52.834) with an efficiency of 93.8% and an average prediction error of 3.7 kg in predicting LW of market/breeding-age pigs reared by smallholder farmers in Mbeya rural and Mbozi districts in the southern highlands zone of Tanzania. It is expected the model will enable more accurate estimation of weight of market/breeding-age pigs in the area. In addition, this study has assessed and confirmed that the available commercially available heart girth tape had reasonable ability to estimate LW of market-age pigs in the area.

## Supporting information

**S1 Dataset.**
(XLSX)

## Acknowledgments

The author wishes to acknowledge all farmers who participated in this study by allowing their pigs to be weighed and measured. The author also thanks all members of the field teams, extension officers and leaders in all study villages.

## Author Contributions

**Conceptualization:** Mwemezi L. Kabululu.

**Data curation:** Mwemezi L. Kabululu.

**Formal analysis:** Mwemezi L. Kabululu.

**Investigation:** Mwemezi L. Kabululu.

**Methodology:** Mwemezi L. Kabululu.

**Project administration:** Mwemezi L. Kabululu.

**Resources:** Mwemezi L. Kabululu.

**Software:** Mwemezi L. Kabululu.

**Validation:** Mwemezi L. Kabululu.

**Visualization:** Mwemezi L. Kabululu.

**Writing – original draft:** Mwemezi L. Kabululu.

**Writing – review & editing:** Mwemezi L. Kabululu.

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
