## [Decision Letter · Decision Letter 0]

8 Aug 2023

PONE-D-23-16971Heart girth best predicts live weights of market-age pigs in TanzaniaPLOS ONE

Dear Dr. Kabululu,

Thank you for submitting your manuscript to PLOS ONE. After careful consideration, we feel that it has merit but does not fully meet PLOS ONE’s publication criteria as it currently stands. Therefore, we invite you to submit a revised version of the manuscript that addresses the points raised during the review process.

ACADEMIC EDITOR:

Dear Dr. Mwemezi L. Kabululu

Thanks for submitting your manuscript for consideration by PLOS ONE. Two researchers reviewed your manuscript and provided comments and questions you need to consider in order to improve your manuscript.

I look forward to receiving your rebuttal response.

We look forward to receiving your revised manuscript.

Kind regards,

Anselme Shyaka, Ph.D

Academic Editor

PLOS ONE

Journal Requirements:

Additional Editor Comment:

Dear Dr. Kabululu,

Thanks for submitting your manuscript for consideration by PLOS ONE. Two researchers reviewed your manuscript and provided comments and questions you need to consider in order to improve your manuscript.

I look forward to receiving your rebuttal response.

Reviewers' comments:

Reviewer's Responses to Questions

**Comments to the Author**

1. Is the manuscript technically sound, and do the data support the conclusions?

Reviewer #1: Yes

Reviewer #2: Yes

2. Has the statistical analysis been performed appropriately and rigorously? 

Reviewer #1: I Don't Know

Reviewer #2: Yes

3. Have the authors made all data underlying the findings in their manuscript fully available?

Reviewer #1: No

Reviewer #2: No

4. Is the manuscript presented in an intelligible fashion and written in standard English?

Reviewer #1: Yes

Reviewer #2: Yes

5. Review Comments to the Author

Reviewer #1: This paper presents an analysis of pig measurements against weight, as a means to better estimate live pig weight int he absence of weigh scales. It is undertaken in Tanzania, and presents what appears as a thorough statistical analysis. This appears sound, but I would suggest that it be looked at by a statistician before the paper progresses.

My main issue is that this is a crowded space of publication, amazingly. A paper not even cited by the author (https://link.springer.com/article/10.1007/s11250-023-03561-z) has effectively already done this study, coming up with a slightly different but entirely analagous result. It is unclear, other than having taken a slightly different approach, how this manuscript builds on that previously done work. Just being specific to Tanzania (and one small region within it) cannot be enough in this context - if the Marshall paper meets the need well, where is the novelty and the added value?

On this basis I reject the paper, but the author may wish to respond with a thorough rebuttal.

Reviewer #2: //journal to confirm if data has been made available (number 3 above)

Line 38: delete -s in ‘involves’

Line 64 -68: it can be clarified that there were two objectives that the study sought to address (then show how each was addressed, and the results)

Line 70: Good to mention that participants willingly consented to participate in the study (were other data collected, besides the measurements?)

Line 93- 105: to make it easier to follow, perhaps have subheadings under 2.3 corresponding to the two objectives – i.e., development of the model, validation of existing weight estimation tape. What informed the sample size of 400? How were the 40 selected, it is mentioned randomly, but not clear how, in the field – was a list of pigs/ households made available before? I suggest we also have a summary description of these 40

Line 107: Can include the version used

Table 1: may be instead of both use ‘all pigs’

Table 2: shouldn’t ‘all pigs’ be in the row below (n=400?)? please indicate, in brackets, the numbers for the other rows (pigs below market age, and market age pigs)

Table 5: please clarify that this is only for the 40 pigs subjected to both weighing, and the tape—the model estimation results, is it for the 40 (not for all pigs)?

References: review all, check number 18, some of the author names are missing

6. PLOS authors have the option to publish the peer review history of their article (what does this mean?). If published, this will include your full peer review and any attached files.

Reviewer #1: No

Reviewer #2: No

---

## [Author Response · Author response to Decision Letter 0]

28 Aug 2023

COMMENT

My main issue is that this is a crowded space of publication, amazingly. A paper not even cited by the author (https://link.springer.com/article/10.1007/s11250-023-03561-z) has effectively already done this study, coming up with a slightly different but entirely analagous result. It is unclear, other than having taken a slightly different approach, how this manuscript builds on that previously done work. Just being specific to Tanzania (and one small region within it) cannot be enough in this context - if the Marshall paper meets the need well, where is the novelty and the added value?

RESPONSE:

The author appreciates the views of the reviewers. However, the following should be noted;

-The very first draft of the paper was written during June to August 2022, way before the Marshall’s paper was published. At the time this paper was submitted to PLOS ONE, the author was not aware of the Marshall’s paper, hence could not have cited it.

-Although this papers bears some similarities with Marshall’s, different approaches have been used. In addition, for this kind of studies, model developed in one area may not be suitable to be used in another area, due to inherent differences between pigs and management practices. Hence a model developed using Ugandan pigs may not be suitable for use in Tanzanian pigs. Hence, findings of this study are still deemed as important to inform tailor-made practice. A point to add is that, the area that the study was carried out is located within the major pig production area in Tanzania with about a half of the pig population; hence the results could more effectively represent the larger pig population

COMMENT:

Line 38: delete -s in ‘involves’

RESPONSE:

Comment is received. However the ‘involves’ refers to the ‘pig marketing’, hence it is deemed as being right

COMMENT:

Line 64 -68: it can be clarified that there were two objectives that the study sought to address (then show how each was addressed, and the results)

RESPONSE:

Clarification has been made, as suggested

COMMENT:

Line 70: Good to mention that participants willingly consented to participate in the study (were other data collected, besides the measurements?) 

RESPONSE:

A sentence has been added, to mention that farmers ’ willingly participated

COMMENT:

Line 93- 105: to make it easier to follow, perhaps have subheadings under 2.3 corresponding to the two objectives – i.e., development of the model, validation of existing weight estimation tape. What informed the sample size of 400? How were the 40 selected, it is mentioned randomly, but not clear how, in the field – was a list of pigs/ households made available before? I suggest we also have a summary description of these 40

RESPONSE:

Description of how the 400 and 40 pigs were obtained (for model development and validation of the existing heart girth tape, respectively) has been improved to make it more clear

COMMENT:

Line 107: Can include the version used

RESPONSE:

A mention of STATA version used has been added

COMMENT:

Table 1: may be instead of both use ‘all pigs’

RESPONSE:

Suggested correction has been made

COMMENT:

Table 2: shouldn’t ‘all pigs’ be in the row below (n=400?)? please indicate, in brackets, the numbers for the other rows (pigs below market age, and market age pigs)

RESPONSE:

Suggested correction has been made

COMMENT:

Table 5: please clarify that this is only for the 40 pigs subjected to both weighing, and the tape—the model estimation results, is it for the 40 (not for all pigs)?

RESPONSE:

Description on how the 40 pigs were obtained and used has been added in the text

COMMENT:

---

## [Decision Letter · Decision Letter 1]

22 Nov 2023

Heart girth best predicts live weights of market-age pigs in Tanzania

PONE-D-23-16971R1

Dear Dr. Kabululu,

We’re pleased to inform you that your manuscript has been judged scientifically suitable for publication and will be formally accepted for publication once it meets all outstanding technical requirements.

Kind regards,

Anselme Shyaka, Ph.D

Academic Editor

PLOS ONE

Additional Editor Comments (optional):

Dear Dr. Kabululu,

I am delighted to inform you that your paper has been accepted for publication by PLOS ONE.

Reviewers' comments:

Reviewer's Responses to Questions

**Comments to the Author**

1. If the authors have adequately addressed your comments raised in a previous round of review and you feel that this manuscript is now acceptable for publication, you may indicate that here to bypass the “Comments to the Author” section, enter your conflict of interest statement in the “Confidential to Editor” section, and submit your "Accept" recommendation.

Reviewer #1: All comments have been addressed

Reviewer #2: All comments have been addressed

2. Is the manuscript technically sound, and do the data support the conclusions?

Reviewer #1: Yes

Reviewer #2: Yes

3. Has the statistical analysis been performed appropriately and rigorously? 

Reviewer #1: Yes

Reviewer #2: Yes

4. Have the authors made all data underlying the findings in their manuscript fully available?

Reviewer #1: Yes

Reviewer #2: Yes

5. Is the manuscript presented in an intelligible fashion and written in standard English?

Reviewer #1: Yes

Reviewer #2: Yes

6. Review Comments to the Author

Reviewer #1: The author has sufficiently - but really only just sufficiently - made revisions to the manuscript that address my comments. Some new papers have been referenced, but there is still a lack of critical comparison between this and the several earlier attempts to do something similar by other authors. As a real contribution to novelty, therefore, this paper remains lacking and in another journal it would probably be rejected again. PLoS One, however, has a policy of only looking at the technical sound-ness of the work, and not its more global context, so I am able to accept this, though somewhat reservedly.

Reviewer #2: No additional comments. The research is relevant and adds more to evidence that is currently available.

7. PLOS authors have the option to publish the peer review history of their article (what does this mean?). If published, this will include your full peer review and any attached files.

Reviewer #1: No

Reviewer #2: No

---

## [Editor Report · Acceptance letter]

24 Nov 2023

PONE-D-23-16971R1 

Heart girth best predicts live weights of market-age pigs in Tanzania 

Dear Dr. Kabululu:

I'm pleased to inform you that your manuscript has been deemed suitable for publication in PLOS ONE. Congratulations! Your manuscript is now with our production department. 

Kind regards, 

on behalf of

Dr. Anselme Shyaka 

Academic Editor

PLOS ONE